# Tropomyosin-Related Kinase Fusions in Gastrointestinal Stromal Tumors

**DOI:** 10.3390/cancers14112659

**Published:** 2022-05-27

**Authors:** Ji Hyun Lee, Su-Jin Shin, Eun-Ah Choe, Jungyoun Kim, Woo Jin Hyung, Hyo Song Kim, Minkyu Jung, Seung-Hoon Beom, Tae Il Kim, Joong Bae Ahn, Hyun Cheol Chung, Sang Joon Shin

**Affiliations:** 1Department of Internal Medicine, Division of Medical Oncology, Eunpyeong St. Mary’s Hospital, The Catholic University of Korea, Seoul 03312, Korea; jhlee8708@gamil.com; 2Department of Pathology, Gangnam Severance Hospital, Yonsei University College of Medicine, Seoul 06273, Korea; charm@yuhs.ac; 3Department of Internal Medicine, Division of Hospital Medicine, Yonsei University College of Medicine, Seoul 03722, Korea; eunahchoe@yuhs.ac; 4Department of Medicine, Yonsei University College of Medicine, Seoul 03722, Korea; jungyounsum@yuhs.ac; 5Songdang Institute for Cancer Research, Yonsei University College of Medicine, Seoul 03722, Korea; 6Department of Surgery, Yonsei University College of Medicine, Seoul 03722, Korea; wjhyung@yuhs.ac; 7Yonsei Cancer Center, Department of Internal Medicine, Division of Medical Oncology, Yonsei University College of Medicine, Seoul 03722, Korea; hyosong77@yuhs.ac (H.S.K.); minkjung@yuhs.ac (M.J.); beomsh@yuhs.ac (S.-H.B.); vvswm513@yuhs.ac (J.B.A.); 8Department of Internal Medicine, Institute of Gastroenterology, Yonsei University College of Medicine, Seoul 03722, Korea; taeilkim@yuhs.ac

**Keywords:** gastrointestinal stromal tumor, KIT, PDGF receptor tyrosine kinase, neurotrophic tyrosine receptor kinase fusion, tropomyosin-related kinase fusion

## Abstract

**Simple Summary:**

The canonical mutations in gastrointestinal stromal tumors (GISTs) are typically activating mutations in *KIT* and platelet-derived growth factor receptor alpha (PDGFRA). Imatinib, the treatment of choice for GISTs, shows a lower response in *KIT/PDGFRA* wild-type GISTs. Neurotrophic tyrosine receptor kinase (*NTRK)* fusion, which can be treated with an NTRK target agent, has been reported in *KIT/PDGFRA* wild-type GISTs, and, therefore, the Yonsei Cancer Center analyzed *NTRK* fusion incidence in *KIT/PDGFRA* wild-type GISTs. At the Yonsei Cancer Center, *NTRK* fusion was confirmed in 16% of cases. Confirmation of *NTRK* fusion in *KIT/PDGFRA* wild-type GISTs provides important information for improving therapeutic outcomes. *NTRK* fusion was confirmed in 16% of *KIT/PDGFRA* wild-type GIST cases at the Yonsei Cancer Center. Confirmation of *NTRK* fusion in *KIT/PDGFRA* wild-type GISTs will improve therapeutic outcomes.

**Abstract:**

The canonical mutations in gastrointestinal stromal tumors (GISTs) are typically activating mutations in *KIT* and platelet-derived growth factor receptor alpha (PDGFRA). GISTs with non-canonical mutations are a heterogeneous group. Here, we examined tropomyosin-related kinase (TRK) fusion in GIST cases without *KIT/PDGFRA* mutations (*KIT/PDGFRA* wild-type (WT) GISTs). We retrospectively analyzed patients who were diagnosed with GISTs at the Yonsei Cancer Center, Severance Hospital, between January 1998 and December 2016. Thirty-one patients with *KIT/PDGFRA* WT GISTs were included in the analysis. TRK expression in tumor samples was assessed by pan-TRK immunohistochemistry (IHC), and the neurotrophic tyrosine receptor kinase (*NTRK*: the gene encoding TRK) rearrangement was analyzed by fluorescence in situ hybridization (FISH). IHC analyses revealed that five cases in this cohort exhibited a weak to moderate TRK expression. *NTRK1* fusions were detected in three tumor samples, and two samples harbored *NTRK3* fusions. The remaining 26 samples did not harbor *NTRK* fusions. Two types of *NTRK* fusions were detected, and the overall *NTRK* fusion frequency in *KIT/PDGFRA* WT GIST cases was 16% (5/31). Our data provide insights into the molecular alterations underpinning *KIT/PDGFRA* WT GISTs. More effort should be devoted to improve methods to identify this distinct disease subtype within the *KIT/PDGFRA* WT GIST group.

## 1. Introduction

Gastrointestinal stromal tumors (GISTs) are the most common mesenchymal tumors of the GI tract, arising from the interstitial Cajal cells [1]. These tumors account for approximately 20% of soft tissue sarcomas, and have an annual incidence of approximately 10 per 1 million individuals [2,3]. Activating mutations in *KIT* and platelet-derived growth factor receptor alpha (PDGFRA) are considered to be the main oncogenic drivers of GISTs; most GISTs (75%) harbor a mutation in *KIT*, and *PDGFRA* mutations occur in 10% to 20% of GISTs [2]. Cases lacking these mutually exclusive mutations are less common, and have been classified as *KIT/PDGFRA* wild-type (WT) GISTs [4]. *KIT/PDGFRA* WT GISTs are a heterogeneous group of different diseases composed of various clinical phenotypes and molecular characteristics [5].

Imatinib mesylate and other small-molecule tyrosine kinase inhibitors (TKIs), such as sunitinib and regorafenib, have demonstrated anticancer activity in GISTs. Therefore, these drugs have been established as part of the protocol for the treatment of GIST patients. Imatinib is the only TKI approved for both the adjuvant and palliative treatment of GISTs, and it is the first-line therapy for unresectable GISTs. Sunitinib is the second-line therapy for patients with imatinib resistance or intolerance, and regorafenib is the third-line therapy for patients with unresectable metastatic GISTs that no longer respond to imatinib and sunitinib [3]. However, patients with advanced GISTs have different responses to imatinib. Advanced WT GISTs have a 0–45% likelihood of responding to imatinib [6]. Additionally, patients with *KIT/PDGFRA* WT GISTs are unlikely to benefit from imatinib treatment [3].

Tropomyosin-related kinase (TRK) A, TRKB, and TRKC are receptor tyrosine kinases encoded by neurotrophic receptor tyrosine kinase (*NTRK)* 1, *NTRK2*, and *NTRK3*, respectively. *TRK* fusions occur in diverse cancers in children and adults [7]. Two studies have reported that the ETS variant transcription factor 6 (*ETV6)-NTRK3* fusion is an actionable target in *KIT/PDGFRA* WT GISTs [8,9]. Recently, the first-generation TRK inhibitors larotrectinib and entrectinib were shown to have marked and durable antitumor activity in patients with *TRK* fusion-positive cancers, regardless of the age of the patient or the tumor type [10,11]. Therefore, it is important to identify which *KIT/PDGFRA* WT GIST patients can benefit from treatment with this type of drug. We investigated the TRK fusion status in patients with *KIT* WT GISTs using immunohistochemistry (IHC) and fluorescence in situ hybridization (FISH) to determine whether we could identify patients with *KIT/PDGFRA* WT GISTs who may be potential candidates for TRK inhibitors.

## 2. Materials and Methods

### 2.1. Patients

We retrospectively analyzed the records of patients who were diagnosed with a GIST at the Yonsei Cancer Center, Severance Hospital, between January 1998 and December 2016. At the time of diagnosis, all cases underwent immunohistochemistry (IHC) for C-kit (CD117), CD34, smooth muscle actin (SMA), and S-100 protein. All 38 KIT and PDGFRA WT GISTs showed diffuse or focal immunoreactivity for C-kit (CD117) and CD34, while the IHC results for SMA and S-100 protein were negative. Distinct histologic features were not observed. Detailed information on patient characteristics was obtained from an electronic medical database. The observation period lasted until November 2019. This study was approved by the Severance Hospital Institutional Review Board (4-2017-0605). The requirement for written informed consent was waived, given the retrospective nature of the study.

### 2.2. Treatment

A total of 38 patients was found to have a *KIT* WT GIST, and 31 patients with sufficient intact tissue samples were included in the analysis. Twenty-eight patients underwent surgery. Eight patients received imatinib as adjuvant therapy, and two patients received neoadjuvant imatinib followed by adjuvant imatinib. Of the three patients who did not undergo surgery, two were treated with imatinib, and one refused treatment. If patients started developing resistance to imatinib, the dose was increased to up to 800 mg/day. If disease progression occurred with the maximum dose of imatinib (800 mg/day), the treatment regimen was changed, if possible.

### 2.3. KIT and PDGFRA Analysis

A commercial sequencing service was used for *KIT* and *PDGFRA* analyses. The lab used Sanger sequencing (direct sequencing). Exons 9, 11, 13, and 17 for *KIT* and exons 12 and 18 for *PDGFRA* were included for mutation analyses.

### 2.4. IHC for TRK Expression

Archival formalin-fixed paraffin-embedded (FFPE) tissues were obtained and, for each case, a block containing a representative lesion was chosen. Tissue sections from these blocks were used for IHC, which was performed using a Ventana XT automated immunohistochemical staining instrument (Ventana Medical Systems, Tucson, AZ, USA) according to the manufacturer’s protocol. Briefly, 4 μM thick sections were deparaffinized using EZ Prep solution (Ventana Medical Systems) and incubated in a CC1 standard solution (pH 8.4 buffer containing Tris/borate/EDTA) for antigen retrieval. Sections were blocked in Inhibitor D (3% H_2_O_2_) for 4 min at 37 °C to block endogenous peroxidases. Slides were incubated with the primary antibody for 24 min at 37 °C, and then with a universal secondary antibody for 8 min at 37 °C. The primary antibody used for IHC staining was anti-pan-TRK (1:100; EPR17341; rabbit monoclonal; Abcam, Cambridge, MA, USA), which detects TRKA, TRKB, and TRKC. The slides were then incubated in streptavidin–horseradish peroxidase for 4 min at 37 °C, followed by a 4 min incubation in the substrate, a solution containing 3,3′-diaminobenzidine tetrahydrochloride and H_2_O_2_. Finally, the slides were counterstained with hematoxylin and a bluing reagent at 37 °C.

The stained slides were evaluated for the proportion and intensity of pan-TRK staining. The intensity of cytoplasmic staining was scored as negative (0), weakly positive (1), moderately positive (2), or strongly positive (3). The proportion of staining was scored based on the percentage of positive cells. The final score was obtained using the formula: 3 × percentage of strongly positive cells + 2 × percentage of moderately positive cells + 1 × percentage of weakly positive cells. The final H-score was a value within the range of 0 to 300.

### 2.5. FISH

All GIST samples were screened using commercially available *NTRK1*, *NTRK2*, and *NTRK3* split FISH probes (Abnova, Taipei, Taiwan), which detect the rearrangements of these genes. The FISH analysis was performed using 2 µM thick FFPE tissue sections. At least 100 nuclei were evaluated by an experienced pathologist for each sample. *NTRK1, NTRK2*, and *NTRK3* with no gene rearrangements showed closely located red and green signals. If more than 15 of 100 nuclei demonstrated separately located red and green signals, this was deemed to indicate *NTRK* rearrangements (i.e., a cutoff value of 15%) [12].

## 3. Results

### 3.1. Patient Characteristics and Treatment

A total of 821 patients was diagnosed with GISTs between January 1998 and December 2016. Thirty-eight patients had *KIT* and *PDGFRA* WT GISTs, and 31 patients had enough intact tissue samples for analysis. Those 31 patients with *KIT* and *PDGFRA* WT GISTs were included in this study. Twenty-eight patients underwent curative resection with or without neoadjuvant and/or adjuvant imatinib, according to the physician’s decision. Three patients had unresectable disease, of whom two received palliative imatinib and one refused any further treatment. The patients’ characteristics are summarized in Table 1. The median age at diagnosis was 56 years (range: 33–92 years); 14 patients (45.2%) were men and 17 (54.8%) were women. The median follow-up duration was 50 months (range: 3–132 months). Four patients died and three patients relapsed after surgery or adjuvant therapy with imatinib. Of the three patients with recurrent disease, one received adjuvant therapy with imatinib.

When comparing the baseline characteristics of patients with *NTRK* fusion and WT patients, there was a statistically significant difference between the two groups. In *NTRK* WT patients, the tumor location was on the stomach in 53.8% of cases, whereas in patients with NTRK fusion, 20% had the tumor in the stomach, 40% in the small bowel, and 40% in the rectum.

### 3.2. TRK Expression in KIT WT GISTs

The expression of TRK in the 31 tumor samples was assessed using pan-TRK IHC. None of the samples exhibited tumor cells with a strong pan-TRK expression, and only five showed a weak to moderate pan-TRK expression (final score: 20–80) (Table 2). Representative images of TRK expression are presented in Figure 1.

### 3.3. NTRK Fusion Analyses

Tumor samples from 31 patients were tested for *NTRK1*, *NTRK2*, and *NTRK3* fusion by FISH analysis. *NTRK1* fusion was detected in three tumor samples, and two samples harbored *NTRK3* fusions. The remaining 26 samples did not harbor *NTRK* fusions. In the five tumor samples harboring *NTRK* fusions, only one also tested weakly positive for TRK expression. Representative images of *NTRK* fusions are presented in Figure 2. The circles indicate *NTRK1* fusions (Figure 2A) and *NTRK3* fusions (Figure 2B).

### 3.4. Characteristics of the Five Patients Harboring NTRK Fusions

The characteristics of the five patients harboring *NTRK* fusions are presented in Table 3. All the patients underwent curative resection. Imatinib was not available as a therapeutic option when patient #1 underwent surgery. The disease recurred as a perineal mass 9 years after surgery, which was confirmed by biopsy. Her physician recommended imatinib as a palliative therapy, but the patient sought a second opinion. She died from disease progression 3 years after relapse, but there were no treatment details in her medical record. Patient #2 was followed up for 6 years with no evidence of disease and stopped seeing his surgeon, but he is alive and still regularly visits his physicians for other health issues at the same hospital. Patient #3 received adjuvant therapy with imatinib for 1 year after the surgery. The disease recurred 1 year and 2 months after the discontinuation of imatinib. Palliative therapy with imatinib was administered immediately after the diagnosis of recurrence and maintained for 4 years and 10 months. However, she died 2 months after the discontinuation of the drug due to her poor performance. Patient #4 received adjuvant therapy with imatinib for 3 years with no evidence of disease, and she is alive and is followed up on a regular basis. Patient #5 was followed up with for 3 years without evidence of disease in the absence of adjuvant therapy. Commercial RNA-based next-generation sequencing (NGS) (Ion Torrent Dx System, Thermo Fisher Scientific, Waltham, MA, USA) was performed using his resected tumor tissue as recommended by his physician, which revealed an *ETV6-NTRK3* fusion. Images of an ETV6-NTRK3 fusion detected by FISH can be found in Appendix A.

## 4. Discussion

This study reports on *NTRK* fusions present in patients with *KIT/PDGFRA* WT GISTs. In total, 16% of the WT GIST patients (5/31) had a TRK expression, and 16% (5/31) tested positive for *NTRK1* (3/31: patients #1, #3, and #4) or *NTRK3* (2/31: patients #2 and #5) fusions. Among the five patients harboring *NTRK* fusions, an *ETV6-NTRK3* chimeric transcript was detected in one patient (patient #5) using commercial RNA-based NGS.

Fusions of the *NTRK* genes have been clinically identified and reported in various cancers [7,13]. Since *TPM3-NTRK1* was first described in a human colorectal carcinoma [14], *NTRK* fusions have been detected in lung cancer [15], papillary thyroid carcinoma [16], colorectal cancer [17], sarcoma (including GIST) [8,9,10], and rare tumors such as secretory breast cancer [18] and mammary analog secretory carcinoma (MASC) [19]. Fusions involving *NTRK1*, *NTRK2*, or *NTRK3* are the most common oncogenic alterations promoting TRK activation, and the mechanisms of *NTRK* gene fusion are remarkably consistent. Typically, intra- or inter-chromosomal rearrangements form hybrid genes, in which the 3′ sequences of *NTRK1*, *NTRK2*, or *NTRK3*, which include the kinase domain, are fused to the 5′ sequences of a different gene. This fusion produces a chimeric oncoprotein resulting in the ligand-dependent constitutive activation of TRK [20]. These fusions are thought to activate the downstream TRK kinase domain and facilitate aberrant TRK signaling via dimerization (e.g., coiled-coil domain, zinc finger domain, or WD domain). However, alternative mechanisms of dimerization or other unknown mechanisms may also exist. We attempted to perform RNA-based NGS with tumor samples obtained from all 31 patients included in this study, but most of the RNA samples did not meet the quality control criteria. We did identify *ETV-NTRK3* fusion in patient #5 (unpublished data), confirming the results from previous NGS. Two different groups have previously reported that the *ETV6-NTRK3* fusion is a targetable alteration in GISTs lacking mutations in the KIT/PDGFRA/RAS pathway genes (e.g., *BRAF, KRAS,* and *NF1*) and SDH deficiencies [8,9]. Brenca et al. suggested that *ETV6-NTRK3* may trigger the insulin-like growth factor 1 receptor signaling cascade and the alternative nuclear insulin receptor substrate 1 pathway to promote the development of GISTs [8]. *ETV6-NTRK3* fusions have also been reported in leukemia, thyroid cancer, pediatric glioma, secretory breast cancer, congenital mesoblastic nephroma, and MASC [21]. *NTRK1* fusions were detected in this study, but we were unable to identify the upstream fusion partners due to the poor quality of RNA for sequencing.

Many clinical trials involving TRK inhibitors to target tumors harboring *NTRK* fusions are underway. A recent trial showed that LOXO-101, a pan-TRK inhibitor also called larotrectinib, exhibited marked and durable efficacy against TRK fusion-positive cancers [10], which led to fast-tracked approval by the Food and Drug Administration. A pooled analysis of three phase 1/2 clinical trials with larotrectinib suggested that TRK fusions define a unique molecular subgroup of advanced solid tumors, against which this drug is highly effective [22]. Entrectinib was also approved for patients with solid tumors harboring *NTRK* fusions. In this study, patient #5′s physician did not administer imatinib as an adjuvant therapy, because he had a *KIT/PDGFRA* WT GIST. If he experiences disease recurrence, a TRK inhibitor or other available drugs targeting NTRK fusion-positive cancers should be administered.

*NTRK* fusion-positive cancers can be grouped into two general categories according to the frequency at which these fusions are detected. In the first category, rare cancer types are highly enriched for *NTRK* fusions, including secretory breast carcinoma, MASC, congenital mesoblastic nephroma, and infantile fibrosarcoma, with a fusion rate of >90%. In the second category, *NTRK* fusions are found at much lower frequencies (5–25% or <5%) in more common cancers such as breast, lung, and colorectal cancers and melanoma. *KIT/PDGFRA* WT GISTs are also among the tumor types that have been shown to harbor *NTRK* fusions with frequencies of 5–25% [20]. In our study, *NTRK* fusions were detected in 16% of *KIT* WT GISTs. Therefore, despite the rarity of *NTRK* fusions in *KIT/PDGFRA* WT GISTs, it is clinically important to identify patients harboring this targetable biomarker. The accurate and efficient identification of GIST patients who are highly likely to benefit from drugs targeting *NTRK* fusions is critical for successful therapy.

There are different testing methods currently available to detect *NTRK* gene fusions in tumor samples. They include IHC, FISH, reverse transcriptase polymerase chain reaction, and NGS using DNA or RNA. Several groups have suggested workflows to detect *NTRK* fusions, which vary depending on the TRK expression and the incidence of *NTRK* fusions [23,24]. FISH is a well-established method that has been used in both clinical trials and clinical practice to test for *NTRK* fusions. Interestingly, Solomon et al. recently proposed a method to triage specimens based on histology and other molecular findings that efficiently identify tumors harboring these treatable oncogenic fusions. FISH, as well as RNA-level fusion testing, were suggested as confirmatory tests [24]. Penault-Llorca et al. [25] reviewed and discussed several testing methods for NTRK fusion, and proposed a testing algorithm to identify patients with TRK fusion cancers. According to the testing algorithm, FISH and NGS could be used as confirmative testing methods with pan-TRK IHC as a screening test, and pan-TRK IHC results might be helpful for selecting the confirmatory test methods, such as a targeted NGS vs. broad NGS panel. They also mentioned that tumors harboring *NTRK3* rearrangements may have a weaker or false negative expression for pan-TRK IHC, and negative results from FISH or pan-TRK IHC should be confirmed by NGS. For tumors with a low incidence of NTRK fusions, such as GISTs, broad NGS testing is preferred in most suggested diagnostic algorithms [26,27,28], although FISH is also recommended as a confirmatory test. We initially planned to use IHC, FISH, and NGS to detect *NTRK* fusions, with the aim of using IHC as a screening method and employing FISH or NGS as confirmatory tests, depending on the circumstances of each patient. However, the RNA samples did not meet the quality control criteria, because they were retrieved from archived tissue that had been stored for years. In addition, there was discordance between the IHC and FISH results in our study.

However, other pan-TRK antibodies resolved this issue in other studies, which suggests the discordance could have been caused by the pan-TRK antibody that we used. Unfortunately, there were insufficient tumor samples to repeat the IHC.

This study had certain limitations, such as its retrospective nature, relatively small sample size, heterogeneity in tumor sampling, discordance among *NTRK* fusion detection methods, and selection bias, since patients were recruited from a single institution in South Korea. Furthermore, the NTRK fusion test was confirmed in GISTs, which were wild-type, not only for KIT and PDGFRA, but also for the BRAF gene [29]; however, the BRAF gene test was not performed in this study. The BRAF wild type was also not tested, which was another limitation of this study.

Nevertheless, our data provide more insights into *KIT/PDGFRA* WT GISTs. The TRK fusion rate of 16% is significant, because TRK is a target with effective targeted therapeutics that have been verified in many recent studies. Mesenchymal tumors driven by *NTRK* fusions are clinically and morphologically heterogeneous. With an increasing number of clinicopathological entities being associated with *NTRK* fusions, the diagnostic and predictive value of the identification of *NTRK* fusions is uncertain. Recently, mesenchymal tumors in the gastrointestinal tract with *NTRK* fusions were described as gastrointestinal stromal tumors (GISTs), but the nosology of such neoplasms remains controversial. Mesenchymal tumors of the gastrointestinal tract with *NTRK* rearrangements are clinically and morphologically heterogeneous, and few, if any, seem related to GISTs [30]. More studies to elucidate the genetic profile of *KIT/PDGFRA* WT GISTs may significantly improve the treatment outcomes of this group of GIST patients.

## 5. Conclusions

In conclusion, our results provided insights into the molecular alterations underpinning *KIT/PDGFRA* WT GISTs. Although *KIT/PDGFRA* WT GISTs are rare, our clinical predictions could aid physicians in identifying patients eligible for *NTRK* fusion screening and therapeutic targeting. More effort should be devoted to improving methods to target this distinct disease subtype within the *KIT/PDGFRA* WT GIST group.

## Figures and Tables

**Figure 1 cancers-14-02659-f001:**
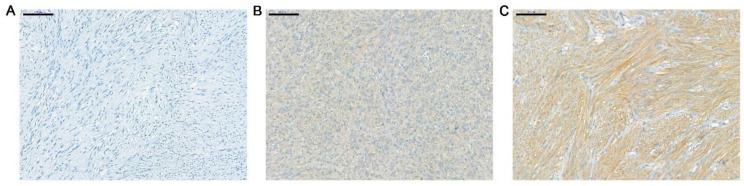
Representative images of pan-TRK expression in samples of WT GIST. IHC results showing samples in which the intensity of cytoplasmic staining was scored as negative (**A**), weakly positive (**B**), and moderately positive (**C**). There were no cases scored as strongly positive. Each scale bar is 100 μM. TRK: tropomyosin-related kinase; WT: wild type; GIST: gastrointestinal stromal tumor; IHC: immunohistochemistry.

**Figure 2 cancers-14-02659-f002:**
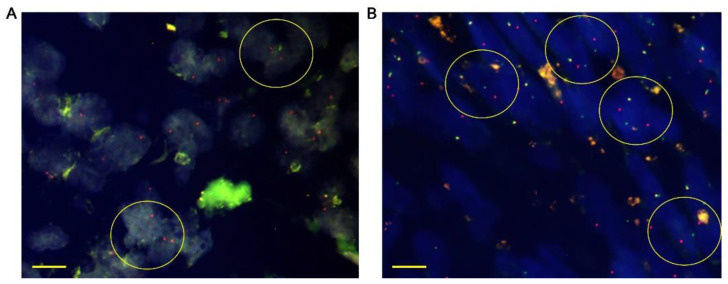
Representative images of *NTRK* fusion detected by FISH. Tumor tissues were stained with dual-color FISH probes. The red and green signals represent upstream and downstream probes, respectively. FISH results showing (**A**) *NTRK1* and (**B**) *NTRK3* fusions. Each scale bar is 10 μM. Circles indicate gene rearrangements. NTRK: neurotrophic tyrosine receptor kinase; FISH: fluorescence in situ hybridization.

**Table 1 cancers-14-02659-t001:** Baseline characteristics.

Characteristics	Total (*n* = 31) *N* (%)	*NTRK* Fusion (*n* = 5)	*NTRK* Wild Type (*n* = 26)	*p*-Value
Age				
Median (range)	56 (33–92)	61 (33–92)	53 (37–84)	0.163
Sex	-	-	-	1000
Male	14 (45.2)	2 (40.0)	12 (46.2)	-
Female	17 (54.8)	3 (60.0)	14 (53.8)	-
Tumor size (cm)	-	-	-	-
Median ± SD	8.2 ± 5.3	3.9 ± 6.5	6.0 ± 5.2	0.410
Tumor location	-	-	-	0.028
Abdominal wall	1 (0.3)	0 (0.0)	1 (3.8)	-
Stomach	14 (45.1)	1 (20.0)	14 (53.8)	-
Small bowel	11 (35.4)	2 (40.0)	6 (23.1)	-
Descending colon	1 (0.3)	0 (0.0)	1 (3.8)	-
Rectum	4 (12.9)	2 (40.0)	4 (15.4)	-
Tumor size, groups (cm) ^◇^	-	-	-	1000
≤5	14 (50.0)	3 (60.0)	11 (47.8)	-
>5	14 (50.0)	2 (40.0)	12 (52.2)	--
Mitotic rate ^◐^ (no. of mitoses/50 HPFs *)		--		0.711
≤5	11 (50.0)	2 (40.0)	12 (52.2)	-
>5	16 (50.0)	3 (60.0)	10 (43.5)	-
Disease status	-	-	-	0.859
No evidence of disease	21 (67.7)	5 (100.0)	16 (61.5)	-
on anticancer treatment	3 (9.7)	0 (0.0)	3 (11.5)	-
Unknown	3 (9.7)	0 (0.0)	3 (11.5)	-
Dead	4 (12.9)	0 (0.0)	4 (15.4)	-
Risk of metastasis ^◈^	-	-	-	0.253
Low	8 (25.8)	1 (20.0)	7 (26.9)	-
Intermediate	3 (9.7)	0 (0.0)	3 (11.5)	-
High	17 (54.8)	4 (80.0)	13 (50.0)	-
Not assessable	3 (9.7)	0 (0.0)	3 (11.5)	-
Surgical treatment	-	-	-	1000
Yes	28 (90.3)	5 (100.0)	23 (88.5)	-
No	3 (9.7)	0 (0.0)	3 (11.5)	-

^◇^ For three inoperable cases, size information and mitotic rate are missing because there was no surgical specimen. ^◐^ Mitotic rate was not evaluated in one surgical specimen. ^◈^ Risk of metastasis was evaluated according to the National Institute of Health (NIH) classification. HPFs: high-power fields; * per 50 HPFs is a total of 5 mm^2^.

**Table 2 cancers-14-02659-t002:** pan-TRK IHC results.

pan-TRK IHC	Intensity of Staining				H-Score
	0	1	2	3	-
Proportion of staining (%)	20%	80%	-	-	80
	20%	80%	-	-	80
	90%	-	10%	-	20
	70%	-	30%	-	60
	70%	-	30%	-	60

IHC: immunohistochemistry; TRK: tropomyosin-related kinase.

**Table 3 cancers-14-02659-t003:** Characteristics of wild-type patients harboring *NTRK* fusions.

Patients (#1–#5)	#1	#2	#3	#4	#5
Age (years)	44	45	65	61	43
Sex	F	M	F	F	M
Location	Rectum	Duodenum	Stomach	Jejunum	Rectum
Mitotic rate (no. of mitoses/50 HPFs *	17	1	70	12	0
Size (cm)	2.8	1.7	17.0	3.9	11.0
Surgery (yes or no)	yes	yes	yes	yes	yes
Follow-up period (years)	11	6	8	4	3
*NTRK* fusion by FISH	*NRTK1*	*NRTK3*	*NRTK1*	*NRTK1*	*NTRK3*
Imatinib	No	No	Adjuvant	Adjuvant	No
Disease/survival status	DDP	NED	DDP	NED	NED

NTRK: neurotrophic tyrosine receptor kinase; FISH: fluorescence in situ hybridization; HPFs: high-power fields; * per 50 HPFs is a total of 5 mm^2^; DDP: dead due to disease progression; NED: no evidence of disease. # Number of patient.

## Data Availability

The data that support the findings of this study are available from the corresponding author upon reasonable request.

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
