# Peer review of "Tropomyosin-Related Kinase Fusions in Gastrointestinal Stromal Tumors"

_cancers, 2022, doi:10.3390/cancers14112659_

Round 1

Reviewer 1 Report

The work of Ji Hyun Lee et al. is focused on the study of gastrointestinal stromal tumor (GISTs) patients in particular with tropomyosin related kinase (TRK) fusion in GIST patients free of KIT/platelet-derived growth factor receptor alpha (PDGFRA) mutations (named as KIT/PDGFRA WT-GISTs). This study combines the use of fluorescence in situ hybridization (FISH) and immunohistochemistry (IHC) assays to visualize the neurotrophic tyrosine receptor kinase (NTRK) fusions in KIT/PDGFRA WT-GIST patients. It has been previously demonstrated the potential capabilities of FISH and IHC techniques for screening and final diagnosis to detect NTRK fusions [Marchiò, C.; Scaltriti, M.; Ladanyi, M.; Iafrate, A.J.; Bibeau, F.; Dietel, M.; Hechtman, J.F.; Troiani, T.; López-Rios, F.; Douillard, J.Y.; et al. Esmo reccomendations on the standard methods to detect ntrk fusions in daily practice and clinical research. Ann. Oncol. Off. J. Eur. Soc. Med. Oncol. 2019, 30, 1417-1427. https://doi.org/10.1093/annonc/mdz204]. For this reason, the methodology used is adequate to achieve the objectives of the carried out research. Nevertheless, I find a lack of novelty since the main conclusions have been previously reported by other groups [Shi, E.; Chmielecki, J.; Tang, C-M.; Wang, K.; Heinrich, M.C.; Kang, G.; Corless, C.K.; Hong, D.; Fero, K.E.; Murphy, J.D.; et al. FGFR1 and NTRK3 actionable alterations in “Wild-Type” gastrointestinal stromal tumors. J. Trans. Med. 2016. 14, 339. https://doi.org/10.1186/s12967-016-1075-6]. Considering the high standard of Cancers journal, I think this paper is not suitable for further publication. I would also like to highlight that the showed results are well-discussed during the main body of the reported manuscript and the scientific paper is well written. For all the aforementioned reasons, I may reject this paper to be published in Cancers journal.

--------

ABSTRACT

Some abbreviations are described in “Last summary” section, but others remain unspecified. Please, add the meaning PDGFR and NTRK (both in line 24).

--------

RESULTS

Result section is well-structured and clearly explained. Nevertheless, some aspects must be improved. Table 1 only shows the mean value of the studied characteristics. Some data are susceptible to be presented with the standard deviation (e.g. tumor size (cm)). Please, introduce SD values for the tumor size (cm) presented in Table 1.

Moreover, scale bars should be added in Figure 2 and Figure S1.

--------

CONCLUSIONS

Conclusions section is mandatory. The submitted manuscript lacks of this section to remark the most relevant outcomes of the conducted work.

--------

BIBLIOGRAPHY

The bibliography is not in the proper format of Cancers journal. Authors must take care of this aspect and deeply revise this section.

Author Response

RESPONSE TO COMMENTS FROM REVIEWER #1

General comments

The work of Ji Hyun Lee et al. is focused on the study of gastrointestinal stromal tumor (GISTs) patients in particular with tropomyosin related kinase (TRK) fusion in GIST patients free of KIT/platelet-derived growth factor receptor alpha (PDGFRA) mutations (named as KIT/PDGFRA WT-GISTs). This study combines the use of fluorescence in situ hybridization (FISH) and immunohistochemistry (IHC) assays to visualize the neurotrophic tyrosine receptor kinase (NTRK) fusions in KIT/PDGFRA WT-GIST patients. It has been previously demonstrated the potential capabilities of FISH and IHC techniques for screening and final diagnosis to detect NTRK fusions [Marchiò, C.; Scaltriti, M.; Ladanyi, M.; Iafrate, A.J.; Bibeau, F.; Dietel, M.; Hechtman, J.F.; Troiani, T.; López-Rios, F.; Douillard, J.Y.; et al. Esmo reccomendations on the standard methods to detect ntrk fusions in daily practice and clinical research. Ann. Oncol. Off. J. Eur. Soc. Med. Oncol201930, 1417-1427. https://doi.org/10.1093/annonc/mdz204]. For this reason, the methodology used is adequate to achieve the objectives of the carried out research. Nevertheless, I find a lack of novelty since the main conclusions have been previously reported by other groups [Shi, E.; Chmielecki, J.; Tang, C-M.; Wang, K.; Heinrich, M.C.; Kang, G.; Corless, C.K.; Hong, D.; Fero, K.E.; Murphy, J.D.; et al. FGFR1 and NTRK3 actionable alterations in “Wild-Type” gastrointestinal stromal tumors. J. Trans. Med201614, 339. https://doi.org/10.1186/s12967-016-1075-6]. Considering the high standard of Cancers journal, I think this paper is not suitable for further publication. I would also like to highlight that the showed results are well-discussed during the main body of the reported manuscript and the scientific paper is well written. For all the aforementioned reasons, I may reject this paper to be published in Cancers journal.

Major comments

COMMENT #1; ABSTRACT

Some abbreviations are described in “Last summary” section, but others remain unspecified. Please, add the meaning PDGFR and NTRK (both in line 24).

RESPONSE #1

Thank you for your comment. The meaning of PDGFR is described in line 23. NTRK abbreviation was described in line 24.

COMMENT #2; RESULTS

Result section is well-structured and clearly explained. Nevertheless, some aspects must be improved. Table 1 only shows the mean value of the studied characteristics. Some data are susceptible to be presented with the standard deviation (e.g. tumor size (cm)). Please, introduce SD values for the tumor size (cm) presented in Table 1. Moreover, scale bars should be added in Figure 2 and Figure S1.

RESPONSE #2

Thank you for your valuable comments. Standard deviation values for the tumor size (cm) were added in Table 1. And scale bars were added in Figure 2 and Figure S1.

COMMENT #3; CONCLUSIONS

Conclusions section is mandatory. The submitted manuscript lacks of this section to remark the most relevant outcomes of the conducted work.

RESPONSE #3

Thank you for your kind comment. Conclusion section was added to article in page 9 after discussion section.

COMMENT #4; BIBLIOGRAPHY

The bibliography is not in the proper format of Cancers journal. Authors must take care of this aspect and deeply revise this section.

RESPONSE #4

Thank you for your comment. The bibliography is change to the ACS style by EndNote program.

Reviewer 2 Report

In this manuscript, Ji Hyun Lee et al. carried a study of tropomyosin related kinase (TRK) fusion in gastrointestinal stromal tumours cases lacking KIT/PDGFRA mutations (wild-type GISTs).

The study included a preliminary immunohistochemical (IHC) evaluation by pan-TRK antibody staining, followed by FISH analysis to detect/confirm NTRK1, NTRK2 or NTRK3 gene rearrangements.

Overall, five of 31 (16%) samples were positive by IHC, while, by FISH analysis, 5 of 31 (16%) samples were positive for NTRK1 or NTRK3 rearrangement. Only one sample was positive both by IHC and FISH.

The results are discussed in the light of current literature.

Comments

There are few published data on NTRK-rearranged GISTs. Therefore, this study may be interesting to shed light on these rare tumours. Moreover, because drugs targeting NTRK kinases are clinically available, the validation of methods to identify potentially treatable tumours is of clinical utility.

One weakness of this study is that the description of pathological and immunophenotypic diagnosis is missing. Authors should include these data. This is particularly important because a recent study (cited, Ref 33 in this manuscript) suggested that the majority of NTRK-rearranged gastrointestinal tumours are distinct from GIST.

Data reported in table 1 disagree with table 3. According to table 1 all NTRK fusion positive tumours were located in the small bowel. In contrast, in table 3 tumour location was stomach for one patient, rectum for two patients, duodenum and jejunum for the other two patients.

Figure 2 A The quality of this picture is really not adequate to demonstrate gene rearrangement. Maybe the image is too dark, but it is really very difficult to see the dots of probes within the circled areas.

At page 7, authors cite supplementary figure 1 as NGS. But supplementary figure 1 is FISH, not NGS. 

Concerning the case (#5) fusion-positive by NGS, authors should provide more information about the method used to obtain this result (e.g. which commercial method? Manufacturer? Product name?).

Author Response

RESPONSE TO COMMENTS FROM REVIEWER #2

General comments

There are few published data on NTRK-rearranged GISTs. Therefore, this study may be interesting to shed light on these rare tumours. Moreover, because drugs targeting NTRK kinases are clinically available, the validation of methods to identify potentially treatable tumours is of clinical utility.

Major comments

COMMENT #1

One weakness of this study is that the description of pathological and immunophenotypic diagnosis is missing. Authors should include these data. This is particularly important because a recent study (cited, Ref 33 in this manuscript) suggested that the majority of NTRK-rearranged gastrointestinal tumours are distinct from GIST.

RESPONSE #1

At the time of diagnosis, all 31 cases performed immunohistochemistry (IHC) for C-kit (CD117), CD34, smooth muscle actin (SMA), S-100 protein. All thirty-eight KIT and PDGFRA WT GISTs showed diffuse or focal immunoreactivity for C-kit (CD117) and CD34, while the IHC results for SMA, and S-100 protein were negative. And any distinct histologic features were not observed. The description of pathological and immunophenotypic diagnosis was added in 2.1 Patients (Materials and methods) section (page 2).

COMMENT #2

Data reported in table 1 disagree with table 3. According to table 1 all NTRK fusion positive tumours were located in the small bowel. In contrast, in table 3 tumour location was stomach for one patient, rectum for two patients, duodenum and jejunum for the other two patients.

RESPONSE #2

Thank you for your kin comment. Locations of tumor has been corrected in Table 1.

COMMENT #3

Figure 2 A The quality of this picture is really not adequate to demonstrate gene rearrangement. Maybe the image is too dark, but it is really very difficult to see the dots of probes within the circled areas.

RESPONSE #3

Thank you for your comment. Figure 2 was replaced with a higher resolution figure.

COMMENT #4

At page 7, authors cite supplementary figure 1 as NGS. But supplementary figure 1 is FISH, not NGS.

RESPONSE #4

Thank you for your kin comment. Supplementary figure 1 shows images of an ETV6-NTRK3 fusion detected by FISH. The above is inserted in line 219.

COMMENT #5

Concerning the case (#5) fusion-positive by NGS, authors should provide more information about the method used to obtain this result (e.g. which commercial method? Manufacturer? Product name?).

RESPONSE #5

We performed NGS using Thermo Fisher Scientific's Ion PGM Dx system. The information is added in text (page 7).

Reviewer 3 Report

The authors provide a study on TRK kinases and their fusion in gastrointestinal stromal tumors. The authors show evidence of fusion in a certain number of patients. The different types of fusions might have potential diagnostic/treatment options.

The authors mentioned that there is no evidence of disease in 21 cases in the table. How were these patients selected? What was the reason for the biopsy? Also, in table 1, how was the risk of metastasis characterized?

For figure 1, IHC, the authors could provide H&E staining, clearly demarcate the tumor area, and then show a magnified view of the IHC staining. Could they indicate the H scores for Figures 1A, B and C.?

Could the authors clarify Figure 2? It isn't easy to interpret and a better figure should be provided indicating tissue area, nucleus and probe signals. What is the fusion being detected? With ETV6? What color is the ETV6 probe? The upstream and downstream probes are related to NTRK or ETV6 genetic locus? Could they include pictures of the WT FISH? How would the NTRK FISH look if there was no fusion?

Could the author clarify what they mean by Wt patient in table 3? Did the patient not have any other genetic mutations in KIT or PDGFRA? Is it possible to look into any publicly available data set to find any correlation between KIT/PDGFRA mutations and NTRK fusion? Are those mutually exclusive and at what frequency does the fusion happen?

Author Response

RESPONSE TO COMMENTS FROM REVIEWER #3

General comments

The authors provide a study on TRK kinases and their fusion in gastrointestinal stromal tumors. The authors show evidence of fusion in a certain number of patients. The different types of fusions might have potential diagnostic/treatment options.

Major comments

COMMENT #1

The authors mentioned that there is no evidence of disease in 21 cases in the table. How were these patients selected? What was the reason for the biopsy? Also, in table 1, how was the risk of metastasis characterized?

RESPONSE #1

The patient population of the study is described in the Materials and Methods section (page 2). We retrospectively analyzed the records of patients who were diagnosed with GIST at the Yonsei Cancer Center, Severance Hospital between January 1998, and December 2016. At the time of diagnosis, all 31 cases performed immunohistochemistry (IHC) for C-kit (CD117), CD34, smooth muscle actin (SMA), S-100 protein. All thirty-eight KIT and PDG-FRA WT GISTs showed diffuse or focal immunoreactivity for C-kit (CD117) and CD34, while the IHC results for SMA, and S-100 protein were negative.

Patients with no evidence of disease means that there has been no recurrence since surgical treatment. And a biopsy was performed for diagnosis. The risk of metastasis was evaluated according to the National Institute of Health (NIH) classification. The risk stratification method is added to footnote of Table 1.

COMMENT #2

For figure 1, IHC, the authors could provide H&E staining, clearly demarcate the tumor area, and then show a magnified view of the IHC staining. Could they indicate the H scores for Figures 1A, B and C.?

RESPONSE #2

These figures represent the intensity of cytoplasmic staining of negative, weakly positive, and moderately positive. The final H-score was obtained using the formula: 3 × percentage of strongly positive cells + 2 × percentage of moderately positive cells + 1 × percentage of weakly positive cells.

COMMENT #3

Could the authors clarify Figure 2? It isn't easy to interpret and a better figure should be provided indicating tissue area, nucleus and probe signals. What is the fusion being detected? With ETV6? What color is the ETV6 probe? The upstream and downstream probes are related to NTRK or ETV6 genetic locus? Could they include pictures of the WT FISH? How would the NTRK FISH look if there was no fusion?

RESPONSE #3

We used commercially available NTRK1, NTRK2 and NTRK3 split FISH probes. These split FISH probes are only used to identify gene splits, and these split FISH probes do not specify which fusion partner gene is.

COMMENT #4

Could the author clarify what they mean by Wt patient in table 3? Did the patient not have any other genetic mutations in KIT or PDGFRA? Is it possible to look into any publicly available data set to find any correlation between KIT/PDGFRA mutations and NTRK fusion? Are those mutually exclusive and at what frequency does the fusion happen

RESPONSE #4

Wild type means that only KIT and PDGFRA were wild. We checked NTRK fusion, KIT mutation, and PDGFRA mutation in TCGA cbioportal. There were total of 38 TCGA lists in which NTRK fusion was confirmed, and when KIT and PDGFRA mutations in each sample were checked, all were wild type. The above contents are summarized in a table. In the TCGA data, it was confirmed that the NTRK fusion and the KIT/PDGFRA mutation were mutually exclusive.

mutation gene

(Total mutations)

NTRK fusion

Wild Type

Mutation(fusion)

NRTK fusion frequency

KIT mutations (314)

314

0

0%

PDGFRA mutations (387)

387

0

0%

mutation gene (total mutations)

KIT mutations

PDGFRA mutations

Wild Type

Mutation

KIT mutation frequency

Wild Type

Mutation

PDGFRA mutation frequency

NTRK1 fusion (14)

14

0

0%

14

0

0%

NTRK2 fusion (6)

6

0

0%

6

0

0%

NTRK3 fusion (18)

18

0

0%

18

0

0%

NTRK fusion (38)

38

0

0%

38

0

0%

Round 2

Reviewer 1 Report

The authors have successfully satisfied my previous requests. For this reason, I warmly recommend the present manuscript for further publication in Cancers journal.

Author Response

We are glad that all of our answers to your questions are satisfactory. I will submit the final manuscript with additional English editing.

Reviewer 2 Report

The revised manuscript has been significantly improved.

There is a major problem about Table 1.

In detail: 

Page 1 line 25: change “which can use” with “which can be treated with”

Page 2 line 47, and page 9 line 322: I suggest to change “enrich” with “identify”.

Page 2 line 90: delete 31. The phrase should become “At the time of diagnosis all cases performed immunohistochemistry…….”

Page 4 line 158: “the tumor location of NTRK fusion patients were all small bowel” This is in contrast with data reported in table 1 and table 3.

Major problem: Table 1: Tumor size (cm) median: Check these numbers. If the NTRK positive are those reported in table 3, it is not possible that median is 6. Do you mean the NTRK positive at IHC?

Page 7 lines 219-221: It would be better to move all the text labelled in yellow (Ion…….figure 1) within the Results Section.

Author Response

Response to reviewer’s comment

COMMENTS #1~#5

#1 Page 1 line 25: change “which can use” with “which can be treated with”

#2 Page 2 line 47, and page 9 line 322: I suggest to change “enrich” with “identify”.

#3 Page 2 line 90: delete 31. The phrase should become “At the time of diagnosis all cases performed immunohistochemistry…….”

#4 Page 4 line 158: “the tumor location of NTRK fusion patients were all small bowel” This is in contrast with data reported in table 1 and table 3.

#5 Page 7 lines 219-221: It would be better to move all the text labelled in yellow (Ion…….figure 1) within the Results Section

RESPONSE

Thank you for your comments. All the above comments are reflected.

COMMENT #6

Major problem: Table 1: Tumor size (cm) median: Check these numbers. If the NTRK positive are those reported in table 3, it is not possible that median is 6. Do you mean the NTRK positive at IHC?

RESPONSE

Thank you for your kind comment. I corrected the median diameter. The median diameter of NTRK Wild type patients was also corrected because there was an error (Table 1).  And we investigated the TRK fusion status in patients with KIT WT GIST using immunohistochemistry (IHC) and fluorescence in situ hybridization (FISH).